# Hereditary Spherocytosis: Linking Ion Transport Defects to Osmotic Gradient Ektacytometry Profiles—A Review

**DOI:** 10.3390/ijms27020721

**Published:** 2026-01-10

**Authors:** Joan Lluís Vives-Corrons, Elena Krishnevskaya

**Affiliations:** Rare Anemias Ektacytometry Unit, Josep Carreras Leukaemia Research Institute, 08916 Barcelona, Spain

**Keywords:** hereditary spherocytosis, ion transport dysregulation, osmotic gradient ektacytometry, red blood cell deformability, erythrocyte membrane disorders, PIEZO1, gardos channel

## Abstract

Hereditary spherocytosis (HS) is the most common inherited red blood cell (RBC) membrane disorder and has traditionally been attributed to defects in cytoskeletal proteins such as spectrin, ankyrin, band 3, and protein 4.2. Growing evidence, however, shows that disturbances in ion transport also contribute to HS pathophysiology. This review summarizes current understanding of HS by integrating membrane structural defects with abnormalities in ion homeostasis and highlights the diagnostic value of osmotic gradient ektacytometry (OGE). Beyond membrane instability, HS erythrocytes exhibit increased cation permeability with abnormal Na^+^ influx and K^+^ loss, leading to cellular dehydration, elevated mean corpuscular hemoglobin concentration (MCHC), and reduced deformability. Dysregulation of mechanosensitive and Ca^2+^-activated K^+^ channels (PIEZO1, KCNN4) may modulate disease expression. OGE—now the reference functional test for RBC deformability—identifies reproducible phenotypes reflecting hydration status, including dehydrated (HS1) and partially hydrated (HS2) HS profiles. When combined with next-generation sequencing (NGS), OGE improves differentiation between HS and overlapping membranopathies such as hereditary xerocytosis or stomatocytosis. In conclusion, HS is a multifactorial disorder resulting from the interplay between cytoskeletal fragility, oxidative stress, and dysregulated ion transport. Integrated diagnostic strategies that combine hematologic indices, OGE, and targeted NGS enhance diagnostic accuracy, support genotype–phenotype interpretation, and guide individualized clinical management. Future efforts should focus on ion-channel modulation and wider adoption of functional assays in precision hematology.

## 1. Introduction

Hereditary spherocytosis (HS) is the most common inherited red blood cell (RBC) membrane disorder in Northern Europe, with an estimated prevalence of 1 in 2000 individuals [1], and remains clinically relevant in the Mediterranean populations [2]. In the Middle East, the estimated prevalence of hereditary spherocytosis ranges from approximately 1 in 2500 individuals. While overall frequency is comparable to that reported in Northern Europe, the disease spectrum differs substantially, with a higher proportion of autosomal recessive and clinically severe forms [2].

Although most cases follow an autosomal dominant inheritance pattern, recessive forms and de novo variants are also well documented [3]. The clinical presentation ranges from compensated mild anemia to severe neonatal hemolysis requiring transfusion or early splenectomy [4], reflecting considerable genetic and phenotypic heterogeneity.

At the molecular level, HS results from quantitative or qualitative defects in membrane and cytoskeletal proteins—most frequently ankyrin (ANK1), band 3 (SLC4A1), β-spectrin (SPTB), α-spectrin (SPTA1), and protein 4.2 (EPB42)—which disrupt the vertical and horizontal interactions that maintain cytoskeletal stability and membrane cohesion [5,6,7] (Figure 1). These abnormalities promote membrane loss through microvesiculation, reduce the surface-to-volume ratio, and transform biconcave erythrocytes into rigid spherocytes that are prematurely cleared by the spleen [8,9,10]. Dominant ANK1, SPTB, and SLC4A1 variants typically produce mild-to-moderate disease, whereas recessive SPTA1 or EPB42 mutations often underlie more severe phenotypes [3,7].

Although HS has historically been considered solely a structural cytoskeletal disorder, longstanding observations of increased cation permeability, abnormal Na^+^ and K^+^ homeostasis, cellular dehydration, and elevated mean corpuscular hemoglobin concentration (MCHC) have highlighted the importance of ion-transport regulation in modulating disease expression [11,12]. These functional abnormalities directly affect RBC deformability and contribute to hemolytic severity, underscoring the need for diagnostic tools capable of assessing membrane mechanics.

Osmotic gradient ektacytometry (OGE), performed with the LoRRca^R^ osmoscan module, has emerged as a key functional assay for quantifying RBC deformability and hydration status [13,14]. Its diagnostic contribution is particularly valuable when traditional tests—such as osmotic fragility, glycerol lysis, EMA binding, or SDS-PAGE electrophoresis—yield inconclusive results [15]. In parallel, next-generation sequencing (NGS) panels targeting genes implicated in hereditary hemolytic anemias have significantly improved diagnostic accuracy and enabled the identification of mixed phenotypes and atypical presentations [16].

Together, OGE and NGS represent complementary tools that support a modern, integrated diagnostic framework for HS—one that incorporates hematologic indices, membrane mechanics, and molecular genetics to refine classification, improve differential diagnosis, and guide individualized clinical management

## 2. Ion Transport Abnormalities in HS

Early studies demonstrated that erythrocytes from patients with HS display increased passive Na^+^ and K^+^ permeability together with compensatory upregulation of the Na^+^/K^+^ ATPase pump [11]. Despite this compensation, HS erythrocytes undergo net Na^+^ gain and K^+^ loss, leading to cellular dehydration and the characteristically elevated MCHC observed in most patients. These disturbances in ion homeostasis contribute to reduced deformability and accelerated hemolysis, but an important conceptual distinction must be made between secondary ion transport abnormalities intrinsic to HS and primary channelopathies that define other HHAs. Specifically, we now report that abnormal Na^+^/K^+^ permeability has been documented in approximately 40–70% of HS patients, depending on disease severity and experimental methodology, and that these abnormalities act as phenotypic modifiers rather than primary etiologic mechanisms.

### 2.1. Secondary Ion-Transport Disturbances Because of Membrane Instability

In HS, most alterations in cation permeability are believed to arise secondary to defects in cytoskeletal proteins, which disrupt the structural organization of the membrane and its associated transporters. Loss of membrane cohesion and changes in lipid–protein interactions may modify the distribution or mechanical environment of ion channels such as the mechanosensitive Ca^2+^ channel PIEZO1 and the Ca^2+^ activated K^+^ channel KCNN4 (Gardos channel) [17,18,19,20,21]. Current evidence suggests that these channels may be functionally influenced by the altered biomechanics of RBCs in HS and not by direct pathogenic mutations in PIEZO1 or KCNN4. Instead, activation of these pathways in HS is thought to reflect mechanotransductive responses to cytoskeletal instability rather than primary channel dysfunction (Figure 2)

### 2.2. PIEZO1/KCNN4 Variants as Phenotypic Modifiers, Not Primary HS Drivers

Although HS itself is not caused by mutations in PIEZO1 or KCNN4, co-inheritance of gain-of-function variants in these genes—classically associated with hereditary xerocytosis (HX)—may modulate RBC hydration and produce a more “dehydrated” HS phenotype. Such combined defects can generate hybrid clinical and ektacytometric patterns, including elevated MCHC and shifts in Ohyper consistent with increased RBC dehydration [22]. This distinction is important: PIEZO1/KCNN4 mutations define primary channelopathies (e.g., HX, Gardos channelopathy), whereas in HS, ion-transport abnormalities primarily reflect downstream biomechanical disruption.

### 2.3. Distinguishing HS from Hereditary Xerocytosis at the Ion-Transport Level

Differentiating HS from HX is clinically essential because splenectomy is contraindicated in HX due to thrombotic risk. Ion-transport physiology provides useful discriminatory features in Hereditary Spherocytosis (HS) and Hereditary Xerocytosis (HX):

HS: Ion leakage is secondary to membrane instability, increased Na^+^ influx and K^+^ loss occur, but Ca^2+^ entry is usually modest, MCHC is elevated but typically within moderate ranges and OGE shows a characteristic pattern with increased Omin and variable Ohyper depending on hydration subtype (HS1 vs. HS2)

HX: Ion-transport defect is primary, most often due to PIEZO1 gain-of-function or KCNN4 mutations [19,23]; chronic Ca^2+^ influx activates the Gardos channel, causing pronounced K^+^ loss and marked cellular dehydration; MCHC can be markedly elevated, often exceeding typical HS values; and OGE is left-shifted with low Ohyper and preserved EImax—distinct from most HS curves [14].

Patch-clamp and ionomic studies support these differences by showing enhanced mechanosensitive channel activity in HX but only subtle or secondary alterations in HS [24]. These physiological distinctions provide important guidance for differential diagnosis and help explain why HS and HX respond differently to splenectomy [25].

Inter-laboratory standardization initiatives have further strengthened the reliability of OGE by establishing reference intervals and quality control protocols. The following section outlines the key conditions that enter the differential diagnosis and highlights how OGE contributes to accurate classification.

## 3. Diagnostic Techniques and Osmotic Gradient Ektacytometry

The diagnosis of HS has traditionally relied on a combination of hematological indices, red cell morphology, osmotic fragility testing, the glycerol lysis test (GLT/AGLT), eosin-5-maleimide (EMA) flow cytometry, and membrane protein electrophoresis by SDS-PAGE [15]. Although these methods remain informative, their sensitivity is limited in mild disease, neonatal presentations, or cases with borderline phenotypic features. Importantly, none of these assays provides a direct assessment of red blood cell (RBC) deformability, a key biophysical determinant of splenic retention and hemolytic severity. To facilitate a structured and reproducible approach to diagnosis, Table 1 summarizes the stepwise diagnostic workflow for hereditary spherocytosis, integrating hematologic indices, confirmatory membrane assays, functional deformability testing by OGE, and targeted next-generation sequencing, along with their characteristic HS findings and clinical interpretive value.

### 3.1. Osmotic Gradient Ektacytometry (OGE): Principles and Parameters

OGE, performed with the Laser-Assisted Optical Rotational Cell Analyzer (LoRRca^R^) and osmoscan module, has emerged as the reference method for functional evaluation of RBC membrane mechanics [13,26,27,28,29]. In this assay, RBCs are exposed to a continuous osmotic gradient while subjected to constant shear stress, and their deformation is quantified through laser diffraction. The resulting “osmoscan” curve plots the Elongation Index (EI) across Osmolarity and provides three essential parameters (Figure 3):Omin: The osmolarity at which EI is minimal, reflecting the point of initial membrane rupture—an indicator of surface-to-volume ratio and hydration.EImax: Maximal RBC deformability, determined mainly by cytoskeletal elasticity and membrane integrity.Ohyper: The osmolarity corresponding to maximal cellular dehydration, influenced by cation content and membrane transport properties.

These parameters offer a quantitative assessment of both membrane structure and cellular hydration status.

OGE results should always be interpreted in conjunction with the reticulocyte count. Significant reticulocytosis may modestly bias deformability upward, but it does not reproduce disease-specific OGE signatures (e.g., hereditary spherocytosis, xerocytosis, or stomatocytosis), which remain diagnostically robust [14,30].

### 3.2. HS Ektacytometric Profiles: HS1 and HS2

OGE has revealed two reproducible RBC deformability phenotypes in HS [14]: HS Type 1 (HS1): This form corresponds to dehydrated, rigid RBCs often associated with high MCHC, increased Omin, decreased Ohyper and reduced EImax and reduced area under the curve. HS Type 2 (HS2): This form reflects a partially hydrated RBC often associated with normal or low MCHC, moderately increased Omin, near-normal Ohyper and mildly reduced EImax (Figure 4)

In our cohort [14], we did not observe a clear correlation between HS1/HS2 ektacytometric patterns and clinical severity, likely due to sample size limitations. However, a subset of the more clinically severe cases appeared within the HS2 group. Importantly, the biological basis of these subtypes may involve differences in secondary ion-transport responses or modifying channel polymorphisms, underscoring the mechanistic interplay between membrane structure and hydration state. These observations require validation in larger series.

### 3.3. OGE in Differential Diagnosis

Beyond HS classification, OGE plays a critical role in distinguishing HS from other hereditary hemolytic anemias. The technique provides disease-specific deformability forms:Hereditary Xerocytosis (HX): HX is characterized by primary ion-transport defects, most often PIEZO1 or KCNN4 gain-of-function mutations, leading to chronic Ca^2+^ entry, Gardos activation, and marked RBC dehydration [19,23]. OGE typically shows a left-shifted curve, with decreased Ohyper, and relatively preserved EImax. Recognition of HX is critical because splenectomy is contraindicated due to a significantly increased risk of thromboembolic events.Hereditary Stomatocytosis (HSt): HSt results from mutations affecting RHAG, SLC4A1, or GLUT1 genes and produces overhydrated RBCs with reduced MCHC. OGE curves are right-shifted, with near-normal or increased Ohyper reflecting excessive cation influx and cell swelling [19].Hereditary Elliptocytosis (HE): HE is caused by defects that impair spectrin self-association or disrupt the horizontal cytoskeletal network. Hydration parameters are usually normal, but EImax is significantly reduced, producing a characteristically flattened trapezoidal-shape OGE curve [31].Congenital Dyserythropoietic Anemia Type II (CDA II). CDA II can mimic HS morphologically but displays ineffective erythropoiesis rather than membrane fragility. OGE curves are normal or only slightly abnormal, aiding exclusion of membranopathies; typically, a normal OGE curve assists in exclusion of membrane pathology [13].

Because misclassification may result in unnecessary or harmful interventions—splenectomy in HX being the prime example—an integrated diagnostic approach combining hematologic indices, OGE patterns, and next-generation sequencing (NGS) is essential (Table 2).

### 3.4. Integration of OGE with Molecular Testing

The combination of OGE with targeted next-generation sequencing (NGS) panels covering 40–50 genes associated with hereditary hemolytic anemias and CDA II, markedly increases diagnostic yield, reaching over 85% in some cohorts [32,33,34,35,36]. OGE provides the functional phenotype, whereas NGS clarifies the molecular basis, enabling precise classification of mixed or atypical cases and supporting genotype–phenotype correlations. Together, these functional and molecular approaches create a robust diagnostic foundation for hereditary red cell disorders. However, the value of OGE becomes most apparent when differentiating HS from other hereditary hemolytic anemias that share overlapping hematologic features but differ markedly in pathophysiology, prognosis, and treatment implications.

Accurate differential diagnosis not only prevents inappropriate interventions—most critically, splenectomy in channelopathies such as hereditary xerocytosis—but also facilitates correct interpretation of genetic findings. Indeed, recognition of the underlying gene defect is essential for predicting disease severity, identifying modifying variants, and understanding the biophysical basis of the observed OGE patterns. Section 4 expands this perspective by examining genotype–phenotype correlations and the molecular mechanisms that shape HS diversity.

## 4. Genotype–Phenotype Correlations and Molecular Insights

More than 500 pathogenic variants associated with HS have been identified across genes encoding membrane structural proteins, most commonly ANK1, SPTB, SLC4A1, SPTA1, and EPB42. Dominant mutations typically result in haploinsufficiency or partial reduction of functional protein, which preserves a substantial proportion of membrane stability and therefore produces milder phenotypes, while recessive or compound heterozygous variants—especially those affecting SPTA1—lead to markedly reduced or dysfunctional protein expression, resulting in more severe membrane instability and early onset disease [36,37,38].

### 4.1. Established Genotype–Phenotype Relationships

Although significant heterogeneity exists, several genotype–phenotype trends are supported by cohort studies:ANK1 and SPTB truncating variants may lead to reduced membrane cohesion, pronounced surface-area loss, and a tendency toward more dehydrated RBCs, sometimes reflected in HS1-like OGE profiles with elevated MCHC [13].SLC4A1 and EPB42 mutations are well-established causes of HS and often result in mild-to-moderate hemolytic phenotypes; OGE usually shows curves compatible with HS; however, although specific associations between these genotypes and partially hydrated HS subtypes or distinct HS2-like OGE patterns have not been systematically demonstrated, these mutations may preserve hydration better and have been linked to partially hydrated phenotypes, compatible with HS2-like OGE curves [14,37].Severe SPTA1 deficiencies (α-spectrin) are common in recessive HS, producing early-onset and transfusion-dependent disease [37,38].

These associations remain probabilistic rather than deterministic, and significant overlap exists.

### 4.2. PIEZO1 and KCNN4 Variants as Modifiers, Not Primary HS Genes

Although HS is not caused by mutations in PIEZO1 or KCNN4, co-inheritance of gain-of-function variants in these cation-channel genes has been shown, in multilocus cases, to modify the clinical and laboratory HS phenotype by increasing RBC dehydration and producing DHS-like OGE profiles [39,40]. Such combined defects may produce hybrid spherocytosis–xerocytosis profiles, reinforcing the importance of joint functional and molecular evaluation.

### 4.3. OGE as a Functional Bridge Between Genotype and Phenotype

OGE provides a reproducible biophysical signature that reflects the combined effects of (a) membrane integrity (affecting EImax), (b) surface-area–to-volume ratio (affecting Omin), and (c) hydration status and ion flux (affecting Ohyper). These biophysical outputs help contextualize genotypes with overlapping or ambiguous clinical presentations by indirectly reflecting surface-to-volume ratio and red cell hydration. Omin is strongly associated with osmotic fragility indices, while Ohyper shows a negative correlation with MCHC, consistent with its role as a marker of intracellular hydration and viscosity [13,41].

### 4.4. Toward an Integrated Diagnostic Approach

The combination of NGS and OGE significantly enhances diagnostic precision and permits recognition of novel or blended phenotypes. Table 3 further illustrates the comparative features of HS subtypes, hereditary xerocytosis, stomatocytosis, elliptocytosis, and CDA II—emphasizing their distinct OGE shifts, membrane or channel defects, associated genes, and clinical implications for splenectomy. Multi-omics approaches—including proteomics and ion-focused analyses—are increasingly used to refine our understanding of how cytoskeletal abnormalities impact membrane transport and red blood cell hydration, supporting a spectrum/continuum model of hereditary red cell membrane disorders [42].

These molecular and biophysical insights have increasingly important clinical consequences. As functional deformability parameters and genetic profiles become more tightly integrated, opportunities arise to refine prognosis, tailor treatment, and improve long-term outcomes. Section 5 explores how this evolving knowledge translates into practical clinical decision-making, with implications for diagnosis, prognostic stratification, splenectomy planning, and the development of emerging therapies

## 5. Clinical and Translational Implications

A clearer understanding of how structural membrane defects interact with ion-transport abnormalities in hereditary spherocytosis (HS) has several important clinical and translational consequences. Increased oxidative damage to membrane lipids and proteins causes oxidative modification of cytoskeletal and anchoring proteins, further weakening membrane stability. Moreover, functional interactions between oxidative stress and ion transport, whereby oxidative membrane damage increases cation leakiness and sensitizes mechanosensitive channels, indirectly exacerbating dehydration and hemolysis.

Integrating hematologic indices, osmotic gradient ektacytometry (OGE), and next-generation sequencing (NGS) allows a more refined assessment of disease severity, prognostic outlook, and therapeutic suitability [30].

### 5.1. Diagnostic Accuracy and Patient Stratification

OGE provides quantitative deformability and hydration metrics that complement traditional assays. When combined with targeted NGS, it enables precise distinction between HS and overlapping membranopathies such as xerocytosis or stomatocytosis—conditions in which therapeutic decisions differ markedly. This functional–molecular integration is especially valuable in borderline cases, mild phenotypes, neonates, or individuals with atypical morphology.

### 5.2. Prognostic Insights

Although the correlation between OGE subtypes (HS1 vs. HS2) and clinical severity requires validation in larger cohorts, certain patterns have practical implications. HS1 phenotype, characterized by RBC dehydration and higher MCHC, is generally associated with reduced deformability and may benefit more from splenectomy. HS2 phenotype shows relatively preserved hydration and may display variable or less predictable responses to splenectomy. These emerging associations support the role of OGE as a potential prognostic biomarker in personalized HS management.

### 5.3. Implications for Splenectomy

Splenectomy remains the principal disease-modifying intervention for moderate to severe HS, but its use must be selective due to long-term risks such as infection and thrombosis. Accurate differentiation from hereditary xerocytosis—where splenectomy is contraindicated—is essential. The combined use of OGE and molecular testing minimizes the risk of misclassification and helps guide safe therapeutic decision-making. Partial splenectomy and endovascular embolization have been explored as intermediate approaches to mitigate complications in selected pediatric cases, though long-term outcomes remain under study.

### 5.4. Emerging Therapeutic Concepts

As understanding of ion transport physiology grows, novel therapeutic targets have been proposed. Gardos channel inhibitors, such as senicapoc, have demonstrated clear efficacy in reducing RBC dehydration in sickle cell disease (SCD) by limiting potassium efflux and improving intracellular hydration [43]. These effects include reduced dense cell fractions, improved hemolytic markers, and increased hemoglobin levels, supporting the biological plausibility of targeting the Gardos channel to modulate RBC hydration. However, no clinical trials or observational studies have evaluated senicapoc—or any Gardos channel inhibitor—in HS. Therefore, any potential benefit in HS must be regarded strictly as a theoretical extrapolation, based on pathophysiological reasoning rather than empirical evidence. While some HS phenotypes exhibit varying degrees of RBC dehydration, it remains unknown whether modulating KCNN4 activity would meaningfully improve hydration, deformability, or clinical outcomes in this disorder. At present, there is no clinical or experimental proof supporting the use of Gardos channel inhibitors in HS, and their role, if any, remains hypothetical. PIEZO1 modulators and Ca^2+^ entry blockers are under active preclinical development and, based on functional studies in human RBCs showing that pharmacological activation or inhibition of PIEZO1 alters Ca^2+^ influx and cell volume, may ultimately offer tools to modify erythrocyte hydration [44,45]. Micronutrient modulation, most notably magnesium supplementation, has been shown to reduce RBC dehydration and improve cation transport abnormalities in SCD [46,47]. These data support the biological plausibility of targeting erythrocyte ion homeostasis through Mg^2+^ supplementation. Although no studies have evaluated this approach in HS, dehydrated HS phenotypes could theoretically benefit from similar strategies, and this warrants future investigation. These approaches remain exploratory, but they highlight the growing therapeutic relevance of functional RBC physiology.

### 5.5. Precision Hematology and Network-Based Approaches

International networks such as ENERCA, EuroBloodNet, and the Rare Anemias International Network (RAIN) support standardization of OGE, coordination of registries, and harmonization of diagnostic criteria. Such collaborative efforts promote improved patient stratification, facilitate multicenter research, and accelerate the adoption of precision hematology in clinical practice [48].

While current approaches already enhance diagnostic precision and patient management, ongoing technological and scientific advances promise to further transform the field. The integration of multi-omics, artificial intelligence–driven curve analysis and standardized global workflows is poised to deepen our understanding of HS pathophysiology and open avenues for personalized treatment. These future directions are discussed in detail in the next section

## 6. Future Directions and Perspectives

Advances in molecular and biophysical technologies are reshaping the understanding of HS and its position within the spectrum of hereditary RBC disorders. Several directions for future research are emerging.

### 6.1. Integrating Multi-Omics to Understand Disease Heterogeneity

Multi-omics approaches—combining genomics, transcriptomics, proteomics, metabolomics, and ionomics—are expected to clarify how structural defects interact with membrane transport pathways to shape RBC hydration, deformability, and lifespan [49]. Such approaches may uncover new modifiers of clinical severity, explain phenotypic overlap with channelopathies, and refine genotype–phenotype correlations.

### 6.2. Refining the Role of Ion-Channel Modulation

Future investigations should define the extent to which PIEZO1 and KCNN4 activity is altered secondary to cytoskeletal instability in HS; determine whether channel modulators can correct dehydration phenotypes; assess longitudinal changes in ion-transport function following splenectomy or emerging therapies.

Targeted pharmacologic modulation of ion flux remains an attractive but still largely theoretical strategy.

### 6.3. Artificial Intelligence and Automated OGE Interpretation

Machine-learning–based analytical tools are increasingly being applied to red blood cell deformability data and have demonstrated high accuracy in distinguishing hereditary hemolytic anemias using rheoscopic and microfluidic platforms [50,51,52,53]. Although similar approaches are only beginning to be explored for osmotic gradient ektacytometry (OGE), the characteristic shape and parameter signatures of OGE curves make them well-suited for automated classification. Such algorithms, once trained on large, curated datasets, have the potential to reduce inter-laboratory variability, improve early recognition of mixed or atypical membranopathy phenotypes, and eventually support point-of-care deformability testing. These applications should thus be viewed as anticipated developments, grounded in emerging evidence but not yet validated in clinical practice.

### 6.4. Standardization and Global Harmonization of Diagnostic Workflows

Efforts to harmonize OGE protocols, establish reference intervals, and develop robust quality-control frameworks will be essential for wider clinical adoption. International registries integrating OGE results with detailed molecular data will allow more accurate benchmarking of disease severity and treatment outcomes.

### 6.5. Opportunities for Personalized Monitoring and Therapy

A future model of HS management may include (a) longitudinal OGE monitoring to track hydration or deformability changes, (b) targeted therapy based on individual ion-transport profiles, (c) risk stratification for splenectomy based on integrated molecular–functional signatures. Such precision-based strategies aim to optimize outcomes while minimizing complications.

Together, these developments highlight an evolving paradigm in which HS is approached through a unified structural–functional lens supported by advanced diagnostics and emerging therapeutic concepts. This broader perspective frames the conclusions of the present review.

## 7. Conclusions

Hereditary spherocytosis (HS) is increasingly recognized as a complex disorder arising from the interplay between structural membrane defects, altered ion homeostasis, and changes in red blood cell deformability. Advances in functional diagnostics, most notably osmotic gradient ektacytometry (OGE), have reshaped the clinical approach to HS by providing quantitative and reproducible measures of hydration status, cytoskeletal integrity, and membrane mechanics. When interpreted alongside targeted next-generation sequencing (t-NGS), these functional insights allow precise distinction between HS and overlapping membranopathies, support emerging genotype–phenotype correlations, and improve diagnostic accuracy across the spectrum of hereditary hemolytic anemias.

This integrative diagnostic framework has direct clinical implications, informing patient selection for splenectomy, clarifying disease severity, and reducing the risk of misclassification—particularly in dehydrating channelopathies where splenectomy is contraindicated. In parallel, growing understanding of ion-channel behavior in RBCs has opened potential avenues for therapeutic innovation, although pharmacologic modulation of PIEZO1, KCNN4, or the Gardos pathway remains exploratory.

Looking forward, multi-omics technologies, artificial intelligence-assisted interpretation of deformability profiles, and international standardization of OGE are poised to refine our understanding of HS heterogeneity and promote individualized management. Together, these advances define a new era of precision hematology in which HS is approached not solely as a structural membranopathy but as a dynamic condition shaped by integrated molecular and biophysical mechanisms.

## Figures and Tables

**Figure 1 ijms-27-00721-f001:**
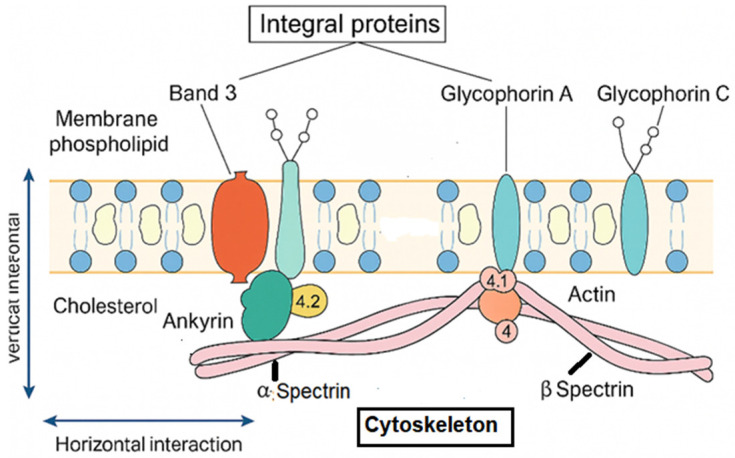
Structural organization of the erythrocyte membrane: vertical and horizontal interactions between integral membrane proteins and the spectrin-based cytoskeleton. Band 3 functions as the major anion exchanger (Cl^−^/HCO_3_^−^) and serves as a key anchoring site linking the lipid bilayer to the cytoskeleton through ankyrin (ANK1) and protein 4.2 (EPB42), thereby ensuring vertical membrane stability. Glycophorin A and C contribute to membrane integrity and provide attachment points for the cytoskeleton via protein 4.1 (EPB41) and actin junctional complexes. The α- and β-spectrin (SPTA1, SPTB) heterodimers form a flexible lattice that confers elasticity, mechanical resilience, and resistance to shear stress through horizontal interactions. Disruption of these structural–functional interactions compromises membrane cohesion, promotes surface-area loss, and underlies the pathophysiology of hereditary spherocytosis.

**Figure 2 ijms-27-00721-f002:**
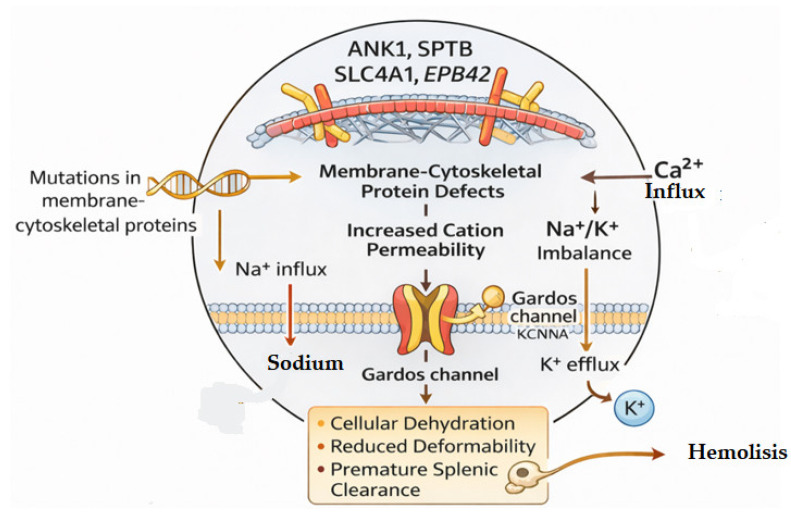
Defects in RBC membrane and cytoskeletal proteins—specifically ANK1, SPTB, SLC4A1, and EPB42—increase cation permeability, causing Na^+^/K^+^ imbalance and Ca^2+^ influx. Elevated intracellular Ca^2+^ activates the Gardos channel, promoting K^+^ efflux and subsequent cell dehydration and reduced deformability.

**Figure 3 ijms-27-00721-f003:**
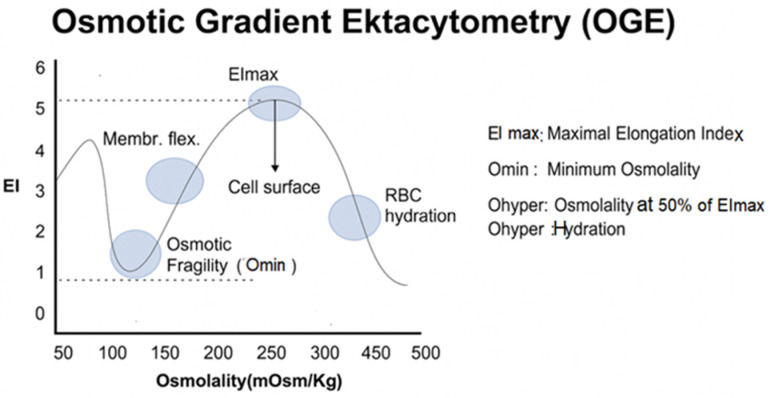
Osmotic Gradient Ektacytometry (OGE) curve showing key biophysical parameters of red blood cells, including EImax (flexibility), Omin (osmotic fragility), and Ohyper (cell hydration state). EI: Elongation Index.

**Figure 4 ijms-27-00721-f004:**
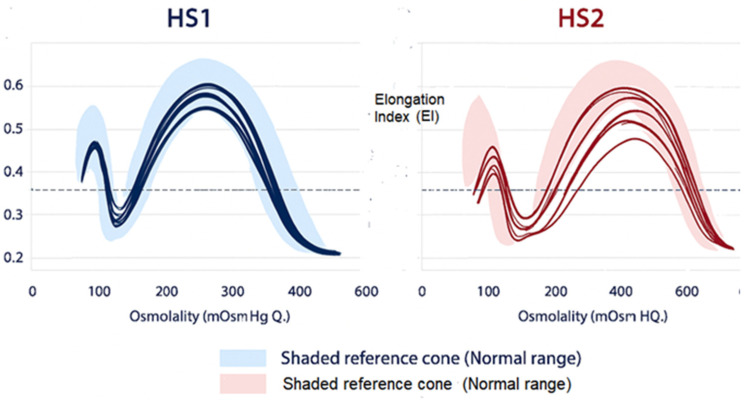
Distinct Osmotic Gradient Ektacytometry (OGE) profiles in Type 1 (HS1) and Type 2 (HS2) Hereditary Spherocytosis showing characteristic alterations in RBC deformability, osmotic fragility, and hydration compared with the normal reference range.

**Table 1 ijms-27-00721-t001:** Diagnostic Workflow Integrating Hematologic, Functional, and Genetic Testing in Hereditary Membranopathies.

Step	Diagnostic Tool	Measured Parameter(s)	Typical HS Finding	Diagnostic Value/Comment
1	Complete blood count (CBC) & blood smear examination	Hb, MCHC, MCV; RBC morphology	↑ MCHC, decreased MCV, presence of spherocytes	Initial screening
2	EMA-binding test (flow cytometry)	Mean fluorescence intensity of band 3	Decreased fluorescence	Sensitive for HS; rapid confirmation
3	Osmotic fragility/cryohemolysis test	Onset of hemolysis under osmotic stress	Increased fragility	Historical Reference Test
4	Osmotic Gradient Ektacytometry (OGE)	Omin, EImax, Ohyper	Type 1 (↑ Omin, ↓ Ohyper) Type 2 (↑ Omin ↑ Ohyper)	Quantifies deformability and hydration
5	Next-generation sequencing (NGS) panel	Pathogenic variants in *ANK1*, *SPTB*, *SLC4A1*, *SPTA1*, *EPB42*, *PIEZO1*, *KCNN4*	Causative or modifying mutation	Definitive genetic diagnosis
6	Integrative interpretation	Combined clinical, morphological, OGE, and genetic data	Subtype and prognosis	Enables personalized management

**Table 2 ijms-27-00721-t002:** Comparative features for the differential diagnosis of Hereditary RBC membranopathies.

Diso Disorder	Main Gene(s)	Mechanism	Hydration	OGE Osmoscan Curve	Treatment
Hereditary Spherocytosis	**HS1**: *ANK1*, *SPTB*, **HS2**: *SLC4A1*, *EPB42*	Cytoskeletal-membrane anchoring failure	Dehydrated Hydrated	Bell-shaped Decreased EImax Increased Omin Variable Ohyper	Splenectomy often beneficial
Hereditary Xerocytosis	*PIEZO1*, *KCNN4*	Cation leak/channelopathy	Dehydrated	Left-shifted Decreased Omin Decreased Ohyper	Splenectomy contraindicated
Hereditary Stomatocytosis	*RHAG*, *SLC4A1*, *GLUT1*	Increased cation influx	Overhydrated	Right-shifted Increased Omin Increased Ohyper	Splenectomy contraindicated
Hereditary Elliptocytosis	*SPTA1*, *SPTB*	Spectrin self-association defect	Normal	Trapezoidal -flat Decreased EImax	Conservative management
Congenital Dyserythropoietic Anemia Type II	*SEC23B*	Dyserythropoiesis	Normal	Often normal	Supportive

**Table 3 ijms-27-00721-t003:** Key RBC membrane and ion-transport proteins implicated in membranopathies, summarizing their category, biological functions and associated gene abnormality.

Category	Protein	Gene	Principal Function	Disorder(s)
Cytoskeletal	α-Spectrin	*SPTA1*	Structural backbone of RBC membrane; provides elasticity and lateral stability	Hereditary spherocytosis (HS), Hereditary elliptocytosis (HE)
	β-Spectrin	*SPTB*	Cross-links spectrin dimers at actin junctions	HS, HE
Anchoring/Linker	Ankyrin-1	*ANK1*	Couples spectrin network to band 3 and lipid bilayer	HS (dominant)
Membrane Transporter	Band 3 (Anion exchanger 1)	*SLC4A1*	Cl^−^/HCO_3_^−^ exchange; CO_2_ transport; membrane stability	HS, Hereditary stomatocytosis (HSt)
Accessory	Protein 4.2	*EPB42*	Stabilizes ankyrin–band 3 complex	HS (recessive)
Channel/Transporter	PIEZO1	*PIEZO1*	Mechanosensitive Ca^2+^ channel activated by shear stress	Hereditary xerocytosis (HX), Dehydrated HS phenotype (HS1)
	KCNN4 (Gardos)	*KCNN4*	Ca^2+^-activated K^+^ efflux channel controlling RBC hydration	Hereditary xerocytosis (HX) Dehydrated HS phenotype (HS1)
	Rh-associated glycoprotein	*RHAG*	Ammonium/gas transport; part of Rh complex	HSt (overhydrated)
Ion Pumps	Na^+^/K^+^-ATPase	*ATP1A1*	Maintains Na^+^/K^+^ gradient across membrane	Secondary changes in HS
Metabolic/ Oxidative	Glucose-6-phosphate dehydrogenase	G6PD	NADPH generation; antioxidant defense	Hemolytic anemia

## Data Availability

No new data were created or analyzed in this study. Data sharing is not applicable to this article.

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
