# Peer review of "Hereditary Spherocytosis: Linking Ion Transport Defects to Osmotic Gradient Ektacytometry Profiles—A Review"

_ijms, 2026, doi:10.3390/ijms27020721_

Round 1
Reviewer 1 Report
Comments and Suggestions for Authors
At the center of the review lies the task of characterizing the methodology for the differential diagnosis of Hereditary Spherocytosis (HS). The modern clinical diagnosis of HS is based on a comprehensive approach: clinical manifestations (anemia, splenomegaly, jaundice), hematological changes in the blood (spherocytes, reticulocytosis), functional tests (osmotic fragility of erythrocytes, EMA test, filterability), as well as genetic testing to identify mutations in membrane protein genes. The key task is to distinguish HS from other hemolytic anemias (e.g., elliptocytosis) using specific tests and to verify the diagnosis molecularly. HS is a membrane pathology, and the authors describe the structure of the erythrocyte membrane and factors affecting deformability in sufficient detail, which lays the groundwork for understanding the principles of the osmotic gradient ektacytometer. The authors structure, according to diagnostic significance, an extensive set of genetic mutations causing transformation of the erythrocyte membrane, from abnormalities of the cytoskeleton and membrane proteins to disturbances in ion homeostasis systems.
The undoubted value of the review is its well-constructed argument for the necessity of a comprehensive methodological strategy for the clinical diagnosis of HS. In addition to routine laboratory methods, the authors justify the inclusion of a functional test based on osmotic gradient ektacytometry (OGE) and genetic analysis in the diagnostic algorithm, which increases diagnostic efficiency.
The manuscript is well-structured, logically organized, and covers the key points of the differential diagnosis of HS from other congenital erythrocyte membranopathies. The importance of differential diagnosis for making irreversible clinical decisions—such as splenectomy—is particularly emphasized.
Illustrations significantly facilitate the perception of the text. Tables effectively summarize the main points and present knowledge in a condensed form. As a suggestion for the authors—based on Table 2, it would be useful to construct a "clinical decision tree"—a good illustration is never superfluous. The authors also outline directions for future research, primarily the implementation of functional assays in precision hematology laboratory practice. The authors have practical experience in using the proposed diagnostic methods, and their expert opinion is valuable.
The file lacks line numbering, which complicates the review and editing process. The authors are advised to carefully read the text once more to correct typos and punctuation errors.
Furthermore, everything stated below pertains to scientific discussion (which a good review should stimulate!) and is not meant to prompt the authors to make changes or additions to the text.
- The clinical course of HS varies in severity from asymptomatic to severe with massive hemolysis. Sometimes moderately expressed HS can be accompanied by other diseases characterized by splenomegaly, for example, infectious mononucleosis. Co-inheritance of other hematological disorders, such as beta-thalassemia or sickle cell anemia (SCA), can also lead to diagnostic difficulties and diverse clinical manifestations. Iron, folate, or vitamin B12 deficiency can exacerbate anemia and mask the laboratory manifestations of HS. Obstructive jaundice alters the lipid composition of the erythrocyte membrane, masking disease manifestations and reducing hemolysis. For example, it is known that HS can be combined with megaloblastic anemia, where morphology will show discocytes and macrocytes, but there will be few spherocytes. Results of osmotic fragility tests can also be nearly normal. Can other diseases (comorbidities) significantly affect the diagnostic algorithm for HS?
- The ektacytometry method is described as highly informative, but even within the context of HS diagnosis, its limitations—for example, the influence of reticulocytosis—are not discussed. It would be desirable to clarify this aspect.
- When screening for mild forms of HS, the osmotic fragility test, according to the literature, has low sensitivity. At the same time, the well-known acidified glycerol lysis test has shown higher sensitivity. Why do the authors not mention it in their work?
The manuscript is recommended for publication in its present form.
Author Response
We thank the reviewer for the constructive feedback and positive assessment of our manuscript which have helped us improve the clarity, completeness, and overall impact of the review. We appreciate the positive feedback regarding the scope and molecular depth of the article. Below we provide a detailed, point‑by‑point response.

Reviewer 2 Report
Comments and Suggestions for Authors
The reviwe article entitled “Hereditary Spherocytosis: Linking Ion Transport Defects to Osmotic Gradient Ektacytometry Profiles — A Review· by Joan Lluís Vives-Corrons and Elena Krishnevskaya presents hereditary spherocytosis (HS) as a multifactorial red blood cell membrane disorder that extends beyond classical cytoskeletal protein defects to include dysregulated ion transport. The review mentions teh integrated diagnostic approaches to enhance genotype–phenotype correlations and personalized clinical management.
The reviwe is well-structured, however, it must include some points:
- Authors should include epidemiological data on the frequency of ion transport abnormalities in HS or their relative contribution compared to cytoskeletal defects.
- Authors should discuss the molecular mechanisms linking ion channels dysregulation to membrane instability and hemolysis.
- Authors should discuss therapuetic strategies or feasible pharmacological candidates to modulate ion channels.
- Authors mention the participation of oxidative stress as part of HS pathophysiology, but supporting evidence and its interaction with ion transport and cytoskeletal damage is missing, please include it.
- In Figure 1, the authors point to a structure that represents actin filaments as the cytoskeleton, and in reality the proteins included in the diagram are part of the cytoskeleton. Please remove the word cytoskeleton.
Author Response

(The authors gave the same response as above.)

Reviewer 3 Report
Comments and Suggestions for Authors
Thank you for the opportunity to review this manuscript. Hereditary Spherocytosis is among the commonest red blood cell disorders. This review is comprehensive and addresses the disease at the molecular level.
I have a few minor comments/suggestions to improve the manuscript's impact.
1- In the introduction; "and remains clinically relevant in Mediterranean populations". Please elaborate further on the prevalence of the disease. The disease is also common in the Middle East regions due to consanguinity. I suggest including slightly more detail on the epidemiology of HS.
2- Figure 1; can be improved as it only shows the structural organization of the erythrocyte membrane, but it can be improved to also indicate the function of these proteins. This will improve the impact of the manuscript
3- Gene names should be italicized to follow the correct nomenclature; for example, the gene name in table 2. I suggest going through the manuscript to make sure.
4- I highly recommend making a figure that illustrates the electrolyte abnormalities in HS. Indicating the membrane protein abnormalities that lead to it.
5- The statement "Dominant mutations typically result in mild to moderate disease, while recessive or compound heterozygous variants—especially those affecting SPTA1—often produce more severe phenotypes" needs more details explaining how the dominant mutations lead to mild to moderate phenotypes.
I have no other concerns regarding this review article.
Author Response

(The authors gave the same response as above.)

Round 2
Reviewer 2 Report
Comments and Suggestions for Authors
Authors have addressed the points raised. One last point that I suggest to the authors is related to Figure 1, in which they point to actin filaments as the cytoskeleton. I consider that the cytoskeleton includes all the submembrane proteins indicated in the figure and not only the actin filaments. Please remove this arrow.
Author Response
Figure 1 has been revised to eliminate the previous misinterpretation of the cytoskeleton as encompassing all submembrane proteins; this has been clearly addressed by removing the incorrect arrow.
Please find enclosed the final version of the manuscript, which includes the amended Figure 1.
